# Clear amniotic fluid aspiration syndrome: A novel description of an old entity

**Pierre-Yves Robillard**[1,2]*, **Francesco Bonsante**[1,2], **Brahim Boumahni**[1], **Pierre Staquet**[1], **Magali Richard**[1], **Julie Guinaud**[1], **Marine Trigolet**[1], **Sandrine Quiviger**[1], **Silvia Iacobelli**[1,2]

1 Service de Néonatologie, Centre Hospitalier Universitaire Sud Réunion, Saint-Pierre Cedex, La Réunion,
2 Centre d'Etudes Périnatales Océan Indien (CEPOI), Centre Hospitalier Universitaire Sud Réunion, Saint-Pierre Cedex, La réunion

* pierre-yves.robillard@chu-reunion.fr, robillard.reunion@wanadoo.fr

**Data Availability Statement:** All relevant data are within the article.

**Funding:** The authors received no specific funding for this work.

## Abstract

### Background

Clear amniotic fluid aspiration syndrome (CAF-AS) is a very rare event occurring in 0.25% of our term clear amniotic fluids deliveries. The study's aims were: 1. to characterize the risk factors and outcomes associated with Clear Amniotic Fluid Aspiration Syndrome and 2. to compare the outcomes of Clear Amniotic Fluid Aspiration to Meconium Aspiration.

### Methods

This was an observational study over a 22-year period in a single level-3 medical center. Compared were parturient/labor characteristics and neonatal outcomes in cases with suspected Clear Amniotic Fluid Aspiration to cases suspected for Meconium Aspiration.

### Results

Out of 79,620 term deliveries there were 66,705 (83.8%) clear amniotic fluids and 12,915 (16.2%) meconium stained amniotic fluid (MSAF). Of neonates born with clear amniotic fluid, 166 (0.25%) were diagnosed with Clear Amniotic Fluid Aspiration Syndrome (CAF-AS), while 202 (15.7%) of those born with MSAF, were diagnosed with aspiration syndrome (MSAF-AS). Both conditions had comparable rates of mild manifestation (67.5% vs 69.2%, p = 0.63). Persistent pulmonary hypertension (PPH) occurred 5 times less in CAF-AS than MSAF-AS (4% vs 20%, OR 0.17, P< 0.0001) Both conditions presented similar rates of surfactant without PPH (11.1% vs 13.4%, p = 0.87). There was 1 postnatal death in CAF-AS vs 10 in MSAF.

### Conclusion

CAF-AS were quantitatively quite similar in terms of need of actual active intervention of the neonatologists in the delivery room (166 vs 202, i.e. in terms of numbers of cases and not prevalence) to MSAF-AS.We identified in these cases two major specific causes: hyperkinetic explosive deliveries in multiparas and long-lasting episodes of maternal hypotension due to epidural/spinal anaesthesia during labor. Out of 140 million births per year in the

**Competing interests:** The authors have declared that no competing interests exist.

world, it should be of concern that 3 million cases are neglected nowadays. Future studies should evaluate if this CAF-AS should benefit from a more active intervention such as immediate endotracheal suction at birth, this clear fluid being very easy to suction.

## Introduction

Meconium-stained amniotic fluid aspiration syndrome MSAF-AS has been particularly well reported for 5 decades as it may lead in newborns to severe respiratory distress syndrome with a high morbidity and mortality rate [1–7]. The terror induced by this syndrome is precisely magnified by the fact that these severe features occur in term neonates. A recent paper by Huang, Zhu and Cheng [7] comparing practices between China and the USA in the past 60 years is probably one of the best recalling exhaustively decades all the debates concerning management of MSAF-AS. International recommendations are very well synthetized in a table. These authors summarize all the neonatal resuscitation's international recommendations: 1987, 1994, 2000, 2005–2006, 2011, 2016, and 2020 (some of them contradicting the preceding). Monfredini et al. [2] also recall very well all these evolutions and controversies [2].

"Clear amniotic fluid aspiration syndrome" CAF-AS does not yet exist in PubMed search. Typing such research gives 56 results ALL of them speaking of meconium staining (because of the root "amniotic fluid") except one: "Suctioning of clear amniotic fluid at birth: A systematic review" [8], but the authors speak only of suctioning the mouth and nose at birth. If we type "meconium-stained amniotic fluid", we have 1188 results.

CAF-AS is a condition never reported before in the literature. We aim to describe associated risk factors and outcomes and compare them to meconium aspiration syndrome.

## Methods

### Definition of exposure and outcomes

In both clear amniotic fluid aspiration syndrome (CAF-AS) and meconium-stained aspiration syndrome (MSAF-AS), we defined the severity of respiratory distress following criteria described by Choi et al. [3] in MFAF-AS:

The criteria needed for the diagnosis of CAF-AS are: severe respiratory distress in the newborn at birth (Silverman score > 3), non-vigorous hypoxic baby needing tracheal intubation, abundant tracheal production of amniotic fluid at endotracheal suction in the delivery room (and careful timing of this production); **b**ilateral crackling lung sounds; **n**eed of oxygen inducing a transfer in the neonatal department (passive oxygen, n-cpap, intubation); and radiological aspects at chest X ray similar to that of MSAF-AS.

Outcomes assessed were: hours or days of oxygenation, hours or days of n-cpap or intubated mechanical ventilation with oxygen, destruction of surfactant, persistent pulmonary hypertension, death.

- <u>Mild aspiration syndrome</u>: 1. a respiratory distress needing less than 48 hours of oxygen support with typical radiological signs (spots in clumps, pulmonary hilum streaks, "humid lung", "Kerley line" in the right lung). 2. In those (a great minority, 0.4%) who have been intubated in the delivery room because of clinical safety necessity (non-vigorous newborn with bad APGAR score and/or with constant bilateral crackling lung sounds at auscultation, bad Silverman score), and a productive endotracheal suction of less than 15 minutes was coined as mild aspiration.

- Severe aspiration syndrome: 1. a respiratory distress needing more than 48 hours of oxygen support requiring assisted mechanical ventilation for more than 48 hours and is often associated with PPH (Persistent Pulmonary Hypertension of the neonate) with typical radiological signs (spots in clumps, pulmonary hilum streaks, "humid lung", "Kerley line" in the right lung). 2. Evident liquid endotracheal suction at the endotracheal intubation (in the delivery room), productive endotracheal suction of more than 15 minutes, plus signs at the chest X-ray.

Primary intubation means intubation at birth directly in the delivery room. Secondary intubation means that the newborn has been intubated in the neonatal NICU department after transfer, therefore minutes or hours after birth.

All pediatrical interventions in the delivery room (intubation or not) were systematically timed by initializing the infant resuscitation table's chronometer to evaluate the duration of intervention and/or the timing of intubation.

## Statistical analyses

Data is presented as numbers and proportions (%) for categorical variables and as mean and standard deviation (SD) for continuous ones. Comparisons between groups were performed by using $\chi^2$-test; odds ratio (OR) with 95% confidence interval (CI) was also calculated. Paired t-test was used for parametric and the Mann-Whitney $U$ test for non-parametric continuous variables. P-values <0.05 were considered statistically significant. Epidemiological data have been recorded and analysed with the software EPI-INFO 7.1.5 (2008, CDC Atlanta, OMS), EPIDATA 3.0 and EPIDATA Analysis V2.2.2.183. Denmark.

## Ethical approval

This study was conducted in accordance with French legislation. As per new French law applicable to trials involving human subjects, a specific approval of an ethics committee is not required for this non-interventional study with anonymized data.

## Results

Out of 79,620 term deliveries there were 66,705 clear amniotic fluids (83.8% of term births); 166 were diagnosed as having presented aspiration syndrome CAF-AS (0.25% of clear amniotic fluids). In the mean time, out of 12,915 meconium stained liquids MSAF (16.2% of term births); 202 were diagnosed as having presented aspiration syndrome MSAF-AS (15.7%. of meconium-stained fluids).

Table 1 depicts the results of all aspiration syndromes (clear and meconium-stained fluids). In both cases 2/3 of cases were considered as "mild" syndromes (mainly management only by n-cpap). Clear fluids (CAF-AS) were clearly less intubated at birth in the delivery room (no intubation OR 2.3, primary intubation OR 0.33, both P < 0.0001) then MSAF-AS. On the other hand, there was a tendency for a secondary intubation in CAF-AS, OR 1.7, p = 009 (in the neonatal department and not at birth in the delivery room).It is of note that there was a higher prevalence of multiparity in the CAFS group while it has long been known that the MAS-AS is predominent in primiparities (p = 0.01). There were less caesarean sections in CAF-AS OR 0.48, p = 0.01 as compared with MSAF-AS. There was higher rates of secondary transfers in the neonatal department in CAF-AS OR 2.9, p < 0.0001, i.e. newborns with respiratory distress syndromes after a long stay at the maternity ward (even hours) before a transfer decision in the neonatal department.

Table 2 analyses all "severe" cases in both CAF-AS and MSAF syndromes. Figs 1 and 2.

**Table 1. A study of 79,620 term births of which 66,705 fluid liquids and 12,915 meconium stained liquids (16.2% of total births).**

| ITEMS in NEWBORNS PRESENTING ASPIRATION SYNDROME | Clear amniotic fluid (CAF) | Meconium stained amniotic fluid (MSAF) | Odds Ratio [95% CI] | P value |
|---|---|---|---|---|
| | N = 66,705 | N = 12,915 | | |
| | (83.8% of term births) | (16.2% of term births) | | |
| | CAF-AS | MSAF-AS | | |
| | (aspiration syndrome) | (aspiration syndrome) | | |
| | N = 166 | N = 202 | | |
| | (0.25% of clear fluid) | (15.6% of MS liquids) | | |
| **Means term (weeks ±SD)** | 39.0 ±1.13 | 39.59 ±1.05 | | P< 0.0001 |
| **Means birthweigt (Kg±SD)** | 3283 ±456 | 3245 ±519 | | P = 0.46 |
| **Type of aspiration** | • 112 Mild (67.5%) | • 141 Mild (69.4%) | | |
| **syndrome (%)** | • 54 Severe (32.5%) | • 61 Severe (30.6%) | | |
| **Intubations** | | | | |
| • **No intubation** | 73 (44.0) | 51 (25.1) | 2.3 [1.5–3.6] | P < 0.0001 |
| • **Primary intubation (delivery room)** | 77 (46,4) | 140 (69.7) | 0.33 [0.25–0.59] | P < 0.0001 |
| • **Secondary intubation** | 16 (9.6) | 12 (6.0) | 1.7 [0.77–3.7] | 0.09 |
| **Abnormal fœtal heart rate (%)** | 60/159 (37.7) | 137/200 (68,5) | 0.28 [0.18–0.43] | P < 0.0001 |
| **Primiparas (%)** | 72 (43.4) | 115 (56.9) | 0.58 [0.38–0.88] | P = 0.01 |
| **Caesarean section (%)** | 40 (24.1) | 81 (40.1) | 0.48 [0.30–0.75] | P = 0.001 |
| **Mode of deliveries in aspiration syndrome** | | | | |
| • **C-section** | 40 (24.1) | 81 (40.1) | | |
| • **Vaginal** | 98 (59.0) | 80 (39.6) | | |
| • **Vaccum, ventouse** | 16 (9.6) | 32 (15,8) | | |
| • **Forceps** | 5 (3.0) | 4 (2.0) | | |
| • **- spatules** | 5 (3.0) | 5 (2.5) | | |
| • **Vaginal breech** | 2 (1.2) | 0 (0) | | |
| **Transfers neonatology** | | | | |
| • **No transfer** | 36 (21.7) | 29 (14.4) | | |
| • **Primary** | 84 (50.6) | 150 (73.4%) | | |
| • **Secondary** | 46 (27.7) | 23 (11.3) | 2.9 [1.7–5.2] | P < 0.0001 |

Again, we retrieve a tendency of a predominance of secondary transfers in neonatology in CAF-AS, OR 1.5, p = 0.17 as well as secondary intubation OR 1.7, p = 0.10. Need of surfactant without evolution towards persistent pulmonary hypertension occurred similarly in CAF-AS than in MSAF-AS (10–13% of severe inhalations). We did experience Persistent pulmonary hypertensions in CAF-AS needing aggressive ventilation, use of NO (Nitric oxygen) inhalation and/ or HFO (High frequency oscillation) in four of our babies, while it happened 5 times more in meconium-stained deliveries (OR 0.17, P< 0.0001).

In our 22 year experience, we had 1 post-neonatal death in CAF-AS vs 10 in MSAF-AS.

## Clinical conditions associated with CAF-AS

The entity CAF-AS being supposed not to exist, nobody was obliged to write down any special observation or suggestions in these kind of records. Therefore, in this paragraph that we will develop inthe discussion. In these non-protocoled observations, we may recall three main entiteis: First, what we may call "acute events" at birth such as difficult and long extraction (forceps, vacuum, spatulas), prolonged fœtal bradycardia (10 mn), shoulder dystocia, abruption placenta, maternal septic shock, fœtal acute anaemia. the two other features which seem to be

**Table 2. A specific study of SEVERE aspiration syndroms.** Comparisons between clear amniotic and meconium-stained fluid aspirations syndromes.

| ITEMS in NEWBORNS PRESENTING SEVERE ASPIRATION SYNDROME | Clear amniotic fluid (CAF) | Meconium stained Amniotic fluids (MSAF) | Odds Ratios [95% CI] | P value |
|---|---|---|---|---|
| | SEVERE CAF-AS (aspiration syndrome) | SEVERE MSAF-AS (aspiration syndrome) | | |
| | N = 54 | N = 61 | | |
| Means term (weeks ±SD) | 39.0 ±1.13 | 39.59 ±1.05 | | |
| Means birthweigt (Kg±SD) | 3283 ±456 | 3245 ±519 | | |
| Intubations | | | | |
| • No intubation | 2 (3.7) # | 5 (8.2) # | | |
| • Primary intubation (delivery room) | 37 (68.5) | 45 (73.8) | | |
| • Secondary intubation | 15 (27.8) | 11 (18.0) | 1.7 | 0.10 |
| Transfers in neonatology | | | | |
| • No transfer | 12 (22.2) & | 6 (11.5) & | | |
| • Primary | 26 (48.1) | 43 (70.5%) | 0.39 [0.18–0.89] | 0.01 |
| • Secondary | 16 (29.6) | 11 (18.0) | 1.5 | 0.17 |
| Abnormal fœtal heart rate (%) | 25/54 (46.3) | 45/61 (73.8) | 0.31 [0.18–0.43] | < 0.0001 |
| Caesrean section (%) | 18 (32.7) | (45.2) | 0.59 [0.28–1.2] | P = 0.17 |
| Primiparas (%) | 21 (38.8) | 27 (44.2) | | P = 0.55 |
| Need of surfactant (DS) without persistent pulmonary hypertension | 6 (11.1) | 8 (13.1) | | 0.87 |
| Persistent pulmonary hypertension (NO, HFO…) | 4 (7.4) | 20 (32.8) | 0.17 [0.04–0.5] | < 0.0001 |
| Neonatal deaths | | | | |
| • No deaths | 53 (98.2) | 50 (82.0) | | |
| • Stillbirths | 0 (0) | 1 (1.6) | | |
| • Neonatal deaths | 1 (1.8) | 10 (16.4) | 0.10 [0.01–0.77] | **0.007** |
| | Death 0–6 days | Deaths 0–6 days: 9 | | |
| | | 7–27 days: 1 | | |

# n-cpap FiO2 ≥ 40%, need of oxygen minimum 3 days

& see details in Table 3

very specific to the CAF-AS are, second, hyperkinetic rapid deliveries in multiparas ("champagne cork" deliveries) and, third, long-lasting (8–10 minutes) episodes of maternal hypotension due to epidural/spinal anaesthesia (occurring even 30 minutes before expulsion).

Table 3 summarizes severe cases who were NOT transferred in a neonatal department staying afterwards with a scope surveillance at the maternity ward, after a cautious and repeated endotracheal suction at birth until complete disappearance of bilateral crackling lung sounds. We classified them as "severe" aspiration syndrome as we could witness clinically productive liquid endotracheal suction during more than 15 minutes. All these cases happened in the early 2000 (no case after 2009) when the previous generation of neonatologists did not hesitate to intubate newborns with severe respiratory distress, (moreover if they had bilateral crackling lung sounds). This behaviour completely disappeared the last decade.

## Discussion

In our 22-year experience, the clear amniotic liquid aspiration syndrome (CAF-AS) was a very rare event in terms of incidence (0.25% of CAF term birth), but CAF births, comprising 84%

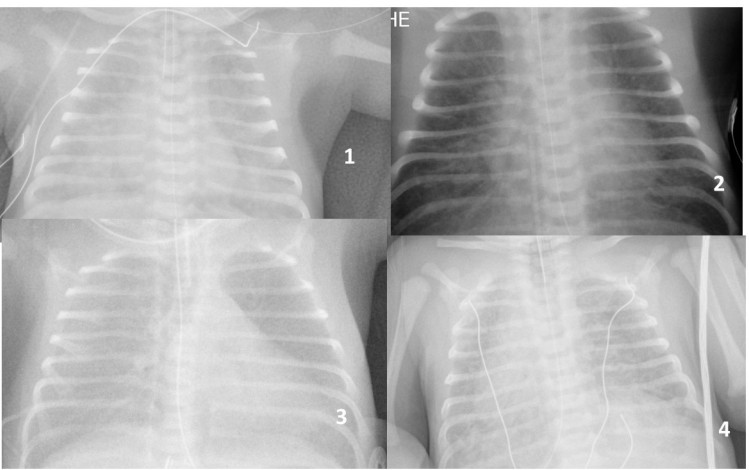

**Fig 1. Clear amniotic fluid (CAF), Intubation delivery room, productive suction more than 15 mn, considered as severe.** 1) crackling lung sounds Intubation delivery room, productive suction 30 mn, 30 hours ventilation. 2) Abruptio placenta, crackling lung sounds, productive endotracheal suction 10 mn, Persistent pulmonary hypertension, NO, 4 days ventilation. 3) Intubation delivery room, productive endotracheal suction 25 mn, 24 hours ventilation. 4) Situs inversus, productive succion 15 mn, FiO2 maximum 40%, extubation day 3.

of births, quantitatively we experienced quite a similar number of lung aspiration syndromes in our term newborns: 202 in meconium-stained amniotic fluids (MSAF) and 166 in clear amniotic fluids (CAF). In daily practice, it is becoming quite similar in neonatologists' interventions in the delivery room, meconium-stained amniotic fluid aspiration syndrome (MSAF-AS) is much more common in terms of incidence (15.7%) among meconium-stained fluids, and presents much more morbidity/mortality than those of clear fluids (see Table 2).

It is of note however that the proportion of lung aspiration syndromes considered mild, see Fig 3 (2/3 of cases, treated mostly by n-cpap only) were similar in both clear and meconium-

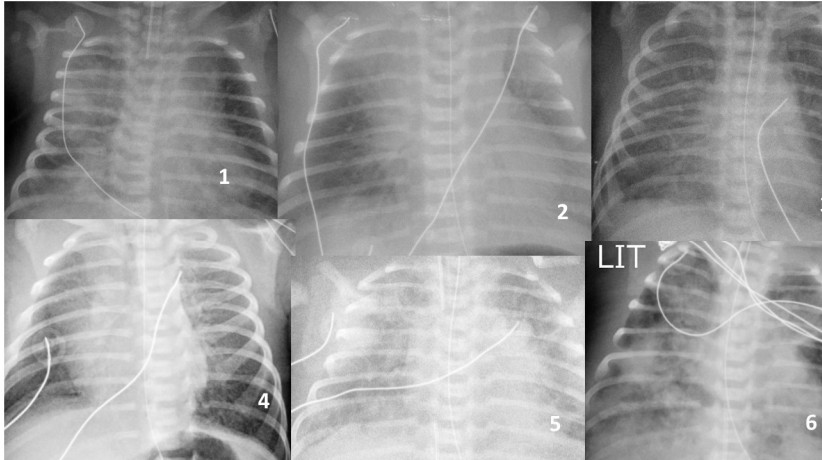

**Fig 2. Clear amniotic fluid (CAF), secondary intubation.** Severe inhalation. 1) RD H2, intubation H2, destruction surfactant, curosurf, ventilation 2 days. 2) Intubation H2, Persistent pulmonary hypertension, HFO, NO, ventilation 3 days. 3) Intubation H3, destruction surfactant, curosurf, ventilation 3 days. 4) RD H3, intubation H21, destruction surfactant, curosurf, ventilation 48 hours. 5) particular case: icthyosis, 4 days ventilation. 6) particular case2: icthyosis, 5 days ventilation.

**Table 3. Details of the newborns with severe lung aspiration syndrome who were not transferred in the neonatal department (scope surveillance in the maternity ward).** N = 18 (12 clear fluid, 6 meconium stained fluid). All intubated at birth in the delivery room.

| ////////////////// | CAF- AS | | MSAF-AS | |
|---|---|---|---|---|
| | Clear amniotic fluid aspiration syndrome | Year of birth | Meconium stained amniotic fluid Aspiration syndrome. | Year of birth |
| Case 1 | Productive suction during 20 mn, | 2003 | | |
| Case 2 | Productive suction during 15 mn, | 2003 | | |
| Case 3 | Productive suction during 20 mn, | 2002 | | |
| Case 4 | Productive suction during 20 mn, | 2004 | | |
| Case 5 | Productive suction during 15 mn, | 2001 | | |
| Case 5 | Productive suction during 25 mn, | 2008 | | |
| Case 6 | Productive suction during 40 mn, | 2001 | | |
| Case 7 | Productive suction during 20 mn, | 2001 | | |
| Case 8 | Productive suction during 30 mn, | 2001 | | |
| Case 9 | Productive suction during 20 mn, | 2005 | | |
| Case 10 | Productive suction during 15 mn, | 2002 | | |
| Case 11 | Productive suction during 20 mn, | 2003 | | |
| Case 12 | Productive suction during 30 mn, | 2003 | | |
| Case 13 | | | Productive suction during 30 mn, | 2004 |
| Case 14 | | | Productive suction during 50 mn, | 2008 |
| Case 15 | | | Productive suction during 35 mn, | 2003 |
| Case 16 | | | Productive suction during 30 mn, | 2001 |
| Case 17 | | | Productive suction during 20 mn, | 2004 |
| Case 18 | | | Productive suction during 20 mn, | 2004 |

stained fluids (Table 1). For the 33% of aspiration syndromes classified as severe, persistent pulmonary hypertension (PPH) occurred 5 times more in MSAF-AS than in CAF-AS (Table 2), and the neonatal mortality rate was also 10 times higher. But, we did have PPH and mortality in CAF-AS, and this pathology (evidence of aspiration syndromes in clear amniotic

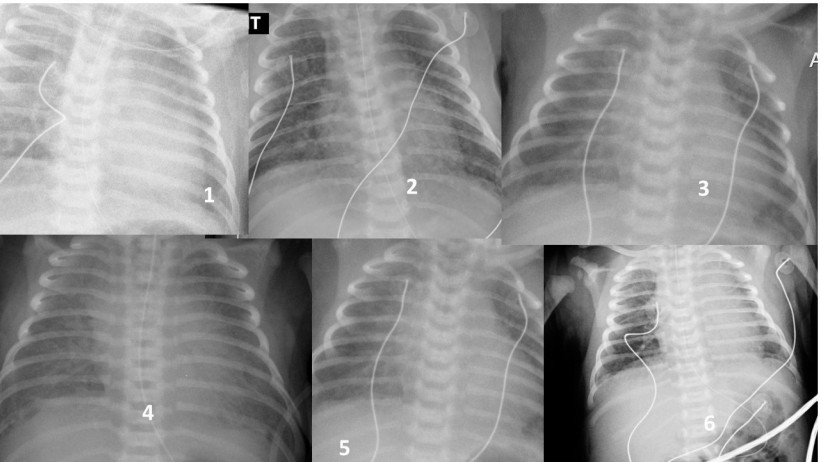

**Fig 3. Clear amniotic fluid (CAF), Inhalation classified as non-severe (treated by n-cpap only).** (babies never intubated). 1) crackling lung sounds, n-cpap 48 hours. 2) RD H2,, n-cpap 4 days (Kerley line). 3) Transfer H3,, n-cpap 2 days. 4) Transfer H1,, n-cpap 2 days. 5) Transfer H2,, crackling lung sounds, n-cpap 36 hours. 6) RD H2, transfer H2, n-cpap 2 days (Kerley line).

fluids deliveries, proven by a productive suction of liquid by endotracheal intubation) has to be absolutely considered.

We did this retrospective observational study to convince the new generation of young neonatologists that this dangerous situation simply exists and that adapted management in the delivery room has to be initialized. It is evident nowadays, in junior-neonatologists' and midwives' minds, that aspiration syndromes may only occur in meconium-stained deliveries. The new generation of both neonatologists and midwives have then been obviously taught in this way. Moreover, there is an indirect piece of evidence in the fact the keyword "CAF-AS, clear amniotic fluid aspiration syndrome" does not exist yet in scientific databases such as Medline searches. Even in the conclusions of their discharge medical letters, these young colleagues make circumvolutions such as "transient respiratory distress", "resorption", "respiratory distress of unknown cause", "neonatal pneumonia". . .. even seeing chest X-rays reported in this paper. They just do not dare to write "clear amniotic fluid aspiration syndrome" (i.e. clearly "humid" lungs at chest X-Ray).

Indirect proofs are evident in the 3 Tables. It is caricatural in Table 3 (newborns classified as having "severe" aspiration syndrome, while remaining afterwards in surveillance at the maternity ward, because of significant clinical productive fluid aspiration from their lungs during dozens of minutes) showing that these kinds of primary intubation did not occur after the year 2009 in our department. Other indirect markers in Tables 1 and 2 suggest relative neglect of these ill newborns: secondary transfers and secondary intubations (in the neonatal department and not at birth) are 3 times higher in CAF-AS than in MSAF-AS. In the case of clear fluid aspiration syndrome in the majority of cases, a neonatologist is not present. He is very often called secondarily by the midwives several minutes (hours?) after birth before a manifest persistent respiratory distress in a newborn (fortunately for these neonates a mild one in 2/3 of cases, see Tables 1 and 2).

## Specificities of CAF-AS, situations of occurrence

In our experience, there are three main features inducing CAF-AS at birth: first, what we may call "acute events" at birth such as difficult and long extraction (forceps, vacuum, spatulas), prolonged foetal bradycardia (10 mn), shoulder dystocia, abruption placenta, maternal septic shock, foetal acute anaemia (1 case, haemoglobin 2. 6g/dl)). We must also recall 8 specific cases encountered on Reunion Island of some familial cases of ichthyosis (see Fig 2) [9, 10].

However, the two other features which seem to be very specific to the CAF-AS are, second, hyperkinetic rapid deliveries in multiparas ("champagne cork" deliveries) and, third, long-lasting (8–10 minutes) episodes of maternal hypotension due to epidural/spinal anaesthesia (occurring even 30 minutes before expulsion). These last cases may increase in the future with the universal diffusion of epidural/spinal anaesthesia everywhere in all countries. It has taken me personally (PYR) two or three decades of interventions in the delivery room to decipher this cause, thanks to an anesthesiologist colleague who told me seeing me fighting with a distressed newborn: "You know, the mother had one hour ago 8 minutes of hypotension, and we had a very hard time to resolve the problem". As a matter of fact, never a neonatologist thinks, before a misunderstood severe case of respiratory distress, to walk afterwards 30 of 40 yards to find the anaesthesiologist asking: "Did something happen during the anaesthesia of the mother?" Coming back to the rapid deliveries in multiparas, it is of note that in Table 1 multiparas are the majority in CAF-AS (p = 0.01), it has been described for long that meconium-stained aspiration syndromes are rather a problem of nulliparas [1]. It seems that in some hyperkinetic labours ("washing machine" effect) the foetus presents in-utero several consecutive gasps leading to inhalation before birth [11, 12].

Among the causes of these 3 main features of CAF-AS, it is difficult to evaluate the actual proportion of each of them, as we have ignored the epidural/spinal/hypotension problem for decades (we can also report one case of maternal hypotension due to an IV injection of antibiotic in an allergic mother). Roughly, we may evaluate between "acute events", rapid deliveries and maternal hypotension with proportions of 30%, 40% and 30%, respectively. Future studies are needed to prospectively refine these evaluations.

## Management in the delivery room of CAF-AS

Besides the current debates in the interest of endotracheal intubation with endotracheal suction of the meconium fluid in the delivery room in MSAF-AC [4, 5, 13, 14] (but recommendations in the MSAF-AS have so many times changed in the last 3 decades, see the "tale" [7]), for example, now in International guidelines for neonatal resuscitation since 2015 suggest against tracheal aspiration of newborns from meconium-stained amniotic fluid deliveries [7]. However, in the current special case of clear fluids we are reporting on, we wish to promote a different debate. Clear liquids being very fluid, they are very easy to suction at birth. We have even many experiences when we perform endotracheal intubation in these babies to have the surprise to face a real initial geyser. Further studies could be to 1) investigate among midwives and anesthesiologists before a misunderstood newborn with breathing effort if something had happened during the long course of labor (in particular, long-lasting hypotensive events in the mother). This would permit to quantify exactly these causes of CAF-AS. 2). In this particular case (clear amniotic fluid), could an immediate endotracheal suction at birth be useful as compared with non-intervention?

## Conclusions

Clear amniotic fluid aspiration syndrome, although a very rare event in incidence (0.25% of term births) is, however, a serious problem as quantitatively it becomes quite similar in neonatologists' interventions in the delivery room to meconium-stained amniotic fluid aspiration syndrome. Among the 140 million births per year in the world, if we extrapolate our experience it concerns some 3 to 3.5 million newborns who are neglected as an existing problem nowadays (very often neonatologists are called dozen of minutes after birth) with possible severe consequences such as secondary need of surfactant and even persistent pulmonary hypertension. It may increase in the future with the universal diffusion of epidural/spinal anaesthesia everywhere. The two major specificities of this CAF-AS are: first, hyperkinetic rapid deliveries in multiparas and, second, long-lasting (8–10 minutes) episodes of maternal hypotension due to epidural/spinal anaesthesia occurring sometimes even one hour before expulsion. Future studies, independent from the 40-year huge debates on management of meconium-stained aspiration syndrome in the delivery room, this specific entity (clear amniotic fluid) may benefit of a specific management at birth. For example, before an evidently "drowned like" baby, severe respiratory distress, bilateral crackling lung sounds, intubation at birth could be considered (these clear liquids being very easy to suction). Finally, the keyword "clear amniotic fluid aspiration syndrome" must be now considered as a valid keyword in all searches of scientific databases.

## Author Contributions

**Conceptualization:** Pierre-Yves Robillard.

**Data curation:** Pierre-Yves Robillard, Brahim Boumahni, Pierre Staquet, Magali Richard, Julie Guinaud, Marine Trigolet, Sandrine Quiviger.

**Formal analysis:** Pierre-Yves Robillard.

**Funding acquisition:** Silvia Iacobelli.

**Investigation:** Pierre-Yves Robillard, Brahim Boumahni, Pierre Staquet, Magali Richard, Julie Guinaud, Marine Trigolet, Sandrine Quiviger.

**Methodology:** Pierre-Yves Robillard.

**Project administration:** Silvia Iacobelli.

**Resources:** Silvia Iacobelli.

**Software:** Pierre-Yves Robillard, Francesco Bonsante.

**Supervision:** Silvia Iacobelli.

**Validation:** Francesco Bonsante, Silvia Iacobelli.

**Writing – original draft:** Pierre-Yves Robillard.

**Writing – review & editing:** Francesco Bonsante, Silvia Iacobelli.

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
