## [Decision Letter · Decision Letter 0]

16 Nov 2023

PONE-D-23-31689

Clear amniotic fluid aspiration syndrome does exist and is quantitatively as frequent as meconium–stained amniotic fluid aspiration syndrome in modern obstetrics. A 22-year experience in Réunion Island.

PLOS ONE

Dear Dr. Robillard,

Thank you for submitting your manuscript to PLOS ONE. After careful consideration, we feel that it has merit but does not fully meet PLOS ONE’s publication criteria as it currently stands. Therefore, we invite you to submit a revised version of the manuscript that addresses the points raised during the review process.

Please respond to all reviewers comments 

We look forward to receiving your revised manuscript.

Kind regards,

Ahmed Mohamed Maged, MD

Academic Editor

PLOS ONE

Journal Requirements:

   "No special fundings besides the normal existence of the perinatal database"

   "No competing interest"

5. Please ensure that you refer to Figures 1 and 3 in your text as, if accepted, production will need this reference to link the reader to the figure.

Reviewers' comments:

Reviewer's Responses to Questions

**Comments to the Author**

1. Is the manuscript technically sound, and do the data support the conclusions?

Reviewer #1: No

Reviewer #2: Yes

2. Has the statistical analysis been performed appropriately and rigorously? 

Reviewer #1: N/A

Reviewer #2: Yes

3. Have the authors made all data underlying the findings in their manuscript fully available?

Reviewer #1: No

Reviewer #2: Yes

4. Is the manuscript presented in an intelligible fashion and written in standard English?

Reviewer #1: No

Reviewer #2: Yes

5. Review Comments to the Author

Reviewer #1: The authors carried out a retrospective study of full-term infants, born between 2001 and 2022, who at birth presented respiratory symptoms and radiological signs possibly suggestive of aspiration of amniotic fluid (clear or meconium-stained). The aim would be to focus the attention of the public on a new and, according to the authors, not uncommon pathology called clear amniotic fluid aspiration syndrome.

Unfortunately, the proposed study has too many methodological limits to be able to reach the fixed objective. All sections of the paper have a lot of limitations. For example, the proposed diagnosis of clear amniotic fluid aspiration syndrome is very questionable; it would require an “… evident liquid suction at the endotracheal intubation (in the delivery room), productive suction of less than 15 minutes…”. Does this mean that all newborns with breathing effort at birth should be intubated to allow a tracheal suction? Intubation can be a simple procedure for experienced physicians but it remains an invasive procedure for the patient, to be performed only if scientific evidence has proven its usefulness. The history of meconium aspiration syndrome tells us that an invasive procedure, such as tracheal aspiration, was introduced into the clinical routine before there was scientific evidence of its usefulness. And yet, such scientific evidence is missing in meconium-stained fluid, and this is the reason why International guidelines for neonatal resuscitation since 2015 suggest against tracheal aspiration of newborns from meconium-stained amniotic fluid deliveries.

The proposed procedure “In our experience, a repeated sequence of [passive oxygenation- cautious thoracic clapping-suction] until the total disappearance of bilateral crackling lung sounds leads to a quasi-recovery of the problem” cannot be acceptable because it lacks scientific foundation and does not take into account the pathophysiology of postnatal respiratory adaptation.

Reviewer #2: The paper investigated an important concept which commonly gives misinformation on clinical practice. As the author concluded , the terminology Clear amniotic fluid aspiration syndrome is not existent and it rarely happens which could be due to expulsive labor in multigravida's. The message which was written on the discussion and conclusion part has to be clear.

6. PLOS authors have the option to publish the peer review history of their article (what does this mean?). If published, this will include your full peer review and any attached files.

Reviewer #1: **Yes: **Simone Pratesi

Reviewer #2: **Yes: **Malede Birara Fanta

---

## [Author Response · Author response to Decision Letter 0]

27 Nov 2023

Reviewers' comments:

Reviewer's Responses to Questions

Comments to the Author

1. Is the manuscript technically sound, and do the data support the conclusions?

Reviewer #1: No

Reviewer #2: Yes

2. Has the statistical analysis been performed appropriately and rigorously?

Reviewer #1: N/A

Reviewer #2: Yes

3. Have the authors made all data underlying the findings in their manuscript fully available?

Reviewer #1: No

Reviewer #2: Yes

4. Is the manuscript presented in an intelligible fashion and written in standard English?

Reviewer #1: No

Reviewer #2: Yes

5. Review Comments to the Author

Reviewer #1: The authors carried out a retrospective study of full-term infants, born between 2001 and 2022, who at birth presented respiratory symptoms and radiological signs possibly suggestive of aspiration of amniotic fluid (clear or meconium-stained). The aim would be to focus the attention of the public on a new and, according to the authors, not uncommon pathology called clear amniotic fluid aspiration syndrome.

PREAMBLE TO THE ANSWERS TO REVIEWER 1. PROFESSOR SIMONE PRATESI. 

Dear colleague, I am (PYR) 69 years old (40 years of NICU) and still « Emeritus Pratician » in my neonatal department in the Maternal and Pediatric Hospital in the University of South-Réunion (a maternity with 4,000 births per year), where I continue to go twice a week to continue my resesarch (Perinatal Epidemiology, preeclampsia, gestational weight gain and fœtal outcomes).

I began my internship on the 1st of October 1980 in the university maternity of Pointe-à-Pitre, Guadeloupe, French West Indies. At this time, the doxa in my department was to intubate ALL newborns (100%) with meconium-stained liquids (and it was the specific duty of interns). We were supposed to follow the Gregory’s famous 1976 study on intubation for thick meconium-stained newborns. During my 4 years of intership, the incidence of MS deliveries was 17% (80% of babies from African-American origin). Therefore, during these 4 years (2,500 births per year there), WE HAVE INTUBATED 1,700 term babies, 90% of them having APGAR 10. As you say perfectly below : « The history of meconium aspiration syndrome tells us that an invasive procedure, such as tracheal aspiration, was introduced into the clinical routine before there was scientific evidence of its usefulness », and I have lived (days and nights) during 4 yeaars this strangeness.

Being intern, I was obliged to obey, and I should have done in 4 years 500 such intubations ( !!) even if did not believe that an APGAR 10 newborn was ill, and as you say also below « Intubation can be a simple procedure for experienced physicians but it remains an invasive procedure for the patient ».

 I became a full practitioner in 1986, and I STOPPED intubating these babies AGAINST THE INTERNATIONAL RECOMMENDATIONS OF THE TIME (see the « TALE » reported by our Chinese colleagues, reference 1. As a matter of fact , in my mind, recommendations concerning meconium-stained births have been the biggest cumulative errors of the young history-40 years- of neonatology). 

Then I went in 1991-1993 to do a 13 months Post-Doc of perinatal epidemiology in Charleston, South-Carolina, USA where they have also a lot of African-American babies. 

The result was :

Alexander GR, Hulsey TC, Robillard PY, De Caunes F, Papiernik E. Determinants of meconium-stained amniotic fluid in term pregnancies. J Perinatol. 1994 Jul-Aug;14(4):259-63. PMID: 7965219.

Where we showed that African newborns had three times more meconium-stained than Caucasian ones only because they were most mature (neglected post-maturity), and not that they were 3 times more “hypoxic stressed”.

I arrived at the University of Réunion in 1999 and established the Perinatal database on January the 1st, 2001. We did as you say « The authors carried out a retrospective study of full-term infants », but all the data have been recorded « au fil de l’eau » prospectively in fact day after day (the coding always done by a neonatologist concerning each newborn, the maternal part filled by a midwife).

In our 23-year database (2001-2023), and I became the chief of the NICU department in 2002 until 2014, the rate of intubation at birth in our term babies :

- MECONIUM-STAINED deliveries (N= 12,999) : TWO percents (2.0%). In my personal professional life, I have lived a rate of 100% intubation in meconium stained deliveries during my internship to 2% when I was Chief of the NICU( !!!).

- Clear amniotic fluid (N= 66,823): ZERO point 4 (0.4%)

But, in these intubated babies, we could also notice that some of them were « drowned » with even a geyser like jet at the intubation, necessitating a minimum of liquid suction (MS as well as clear amniotic fluids). Our work is to give our clinical experience and report that aspiration may also exist in clear liquids.

Biography. Neonatologist, epidemiologist, specialist- in tropical diseases and perinatal epidemiology. 44 years of work in tropical countries (French overseas departments and Territories): 16 years in Guadeloupe (1979-1995, French West-Indies, Caribbean’s), 3 years in Tahiti (1995-1998, French Polynesia, Pacific) and 24 years (since 1999) in Reunion, French overseas Department, Indian Ocean. All his career in level 3 NICU’s and university hospitals. Twelve years (2002-2014) at the head of the neonatal department and NICU of the University hospital South Reunion (Saint-Pierre). Currently working at Centre Hospitalier Universtaire Sud-Réunion, Saint-Pierre.

Creator of the Reunion perinatal epidemiological database (2001), and the International Workshop on Immunology of Preeclampsia (since 1998), and co-founder of the Centre d’Epidémiologie périnatale Océan Indien CEPOI (2010). One year of post-Doctoral Fellowship in perinatal epidemiology (MUSC, Medical University of South Carolina, Charleston, USA, 1991-1992). International course of Epidemiology CDC Atlanta (1992). 

ANSWERS TO REVIEWER 1.

Unfortunately, the proposed study has too many methodological limits to be able to reach the fixed objective.

We have chosen to compare our experience Clear amniotic fluid aspiration with the worst situation : meconium fluid aspiration in term babies to show that our described situation is also (while much less) a significant problem. This « syndrome » being completely neglected, very often these ditressed newborns are fighting for breathing in thircraddle in the maernity ward hours after birth before finding any help.

 All sections of the paper have a lot of limitations. For example, the proposed diagnosis of clear amniotic fluid aspiration syndrome is very questionable; it would require an “… evident liquid suction at the endotracheal intubation (in the delivery room), productive suction of less than 15 minutes…”. 

We have added in Methods, page 4 :

“In those (a great minority, 0.4%) who have been intubated in the delivery room because of clinical safety necessity (non-vigorous newborn with bad APGAR score and/or with constant bilateral crackling lung sounds at auscultation, severe Silverman score) a productive suction of less than 15 minutes was coined as mild aspiration.

Does this mean that all newb »orns with breathing effort at birth should be intubated to allow a tracheal suction? Intubation can be a simple procedure for experienced physicians but it remains an invasive procedure for the patient,

Not at all Sir, in our maternity, only 2.0% of our meconium-stained newborns are intubated in the delivery room and 0.4% of our clear amniotic fluids. This is why we have added in the sentence :

“non-vigorous newborn with bad APGAR score and/or with constant bilateral crackling lung sounds at auscultation, severe Silverman score”

 to be performed only if scientific evidence has proven its usefulness. The history of meconium aspiration syndrome tells us that an invasive procedure, such as tracheal aspiration, was introduced into the clinical routine before there was scientific evidence of its usefulness. 

We totally agree with you. We did not perform because of proven scientific evidence. All the intubated babies were

« In those (a great minority, 0.4%) … have been intubated in the delivery room because of clinical safety necessity”

And yet, such scientific evidence is missing in meconium-stained fluid, and this is the reason why International guidelines for neonatal resuscitation since 2015 suggest against tracheal aspiration of newborns from meconium-stained amniotic fluid deliveries.

We have rewritten page 13 :

Management in the delivery room of CAF-AS. Besides the current debates in the interest of endotracheal intubation with suction of the meconium fluid in the delivery room in MSAF-AC [10-14] (but recommendations in the MSAF-AS have so many times changed in the last 3 decades, see the “tale” [1]), for example, now in International guidelines for neonatal resuscitation since 2015 suggest against tracheal aspiration of newborns from meconium-stained amniotic fluid deliveries [1]. However, in the current special case of clear fluids we are reporting, we wish to promote a different debate. Clear liquids being very fluid, they are very easy to suction at birth. We have even many experiences when we perform endotracheal intubation in these babies to have the surprise to face a real initial geyser. . Further studies could be to 1) investigate among midwives and anesthesiologists before a misunderstood newborns with breathing effort if something happened during the long course of labor (in particular long-lasting hypotensive events in the mother). This would permit to quantify exactly this cause of CAF-AS 2) In this particular case (clear amniotic fluid), could an immediate suction at birth be useful as compared with non-intervention ?

The proposed procedure “In our experience, a repeated sequence of [passive oxygenation- cautious thoracic clapping-suction] until the total disappearance of bilateral crackling lung sounds leads to a quasi-recovery of the problem” cannot be acceptable because it lacks scientific foundation and does not take into account the pathophysiology of postnatal respiratory adaptation.

WE HAVE COMPLETELY DELETED THE SENTENCE :

“In our experience, a repeated sequence of [passive oxygenation- cautious thoracic clapping-suction] until the total disappearance of bilateral crackling lung sounds leads to a quasi-recovery of the problem”

Reviewer #2: The paper investigated an important concept which commonly gives misinformation on clinical practice. As the author concluded , the terminology Clear amniotic fluid aspiration syndrome is not existent and it rarely happens which could be due to expulsive labor in multigravida's. The message which was written on the discussion and conclusion part has to be clear.

We have rewritten the conclusions :

“The two major specificities of this CAF-AS are, first, hyperkinetic explosive deliveries in multiparas and, second, long-lasting (8-10 minutes) episodes of maternal hypotension due to epidural/spinal anaesthesia occurring sometimes even one hour before expulsion. Future studies, independent from the 40-year huge debates on management of meconium-stained aspiration syndrome in the delivery room, this specific entity (clear amniotic fluid) may benefit of a specific management at birth. For example, before an evidently “drowned like” baby, severe respiratory distress, bilateral crackling lung sounds, intubation at birth could be considered (these clear liquids being very easy to suction).”

We thank the reviewers for their positive critics.

---

## [Decision Letter · Decision Letter 1]

25 Jan 2024

PONE-D-23-31689R1Clear amniotic fluid aspiration syndrome does exist and is quantitatively as frequent as meconium–stained amniotic fluid aspiration syndrome in modern obstetrics. A 22-year experience in Réunion Island.PLOS ONE

Dear Dr. Robillard,

Thank you for submitting your manuscript to PLOS ONE. After careful consideration, we feel that it has merit but does not fully meet PLOS ONE’s publication criteria as it currently stands. Therefore, we invite you to submit a revised version of the manuscript that addresses the points raised during the review process.

**Please respond to all reviewers comments**

We look forward to receiving your revised manuscript.

Kind regards,

Ahmed Mohamed Maged, MD

Academic Editor

PLOS ONE

Reviewers' comments:

Reviewer's Responses to Questions

**Comments to the Author**

1. If the authors have adequately addressed your comments raised in a previous round of review and you feel that this manuscript is now acceptable for publication, you may indicate that here to bypass the “Comments to the Author” section, enter your conflict of interest statement in the “Confidential to Editor” section, and submit your "Accept" recommendation.

Reviewer #1: (No Response)

Reviewer #3: (No Response)

Reviewer #4: (No Response)

2. Is the manuscript technically sound, and do the data support the conclusions?

Reviewer #1: (No Response)

Reviewer #3: (No Response)

Reviewer #4: No

3. Has the statistical analysis been performed appropriately and rigorously? 

Reviewer #1: (No Response)

Reviewer #3: (No Response)

Reviewer #4: Yes

4. Have the authors made all data underlying the findings in their manuscript fully available?

Reviewer #1: (No Response)

Reviewer #3: (No Response)

Reviewer #4: No

5. Is the manuscript presented in an intelligible fashion and written in standard English?

Reviewer #1: (No Response)

Reviewer #3: (No Response)

Reviewer #4: No

6. Review Comments to the Author

Reviewer #1: Dear Authors I am very sorry but the modified version of your paper still remain not sufficient to be published, above all in a journal with a high impact factor

Reviewer #3: (No Response)

Reviewer #4: I am very enthusiastic for the manuscript since this is an important entity which neonatologists are aware of but hasn’t been officially reported. Therefore the subject is both novel and clinically important. However, in my opinion it needs professional scientific editing to be taken seriously by readers. I will lay out my suggestions for the abstract to make my point.

7. PLOS authors have the option to publish the peer review history of their article (what does this mean?). If published, this will include your full peer review and any attached files.

Reviewer #1: No

Reviewer #3: No

Reviewer #4: **Yes: **Tal Weissbach

---

## [Author Response · Author response to Decision Letter 1]

8 Feb 2024

DETAILED REVIEW (Reviewer 4). Modifications in red in the text

General comment- I am very enthusiastic for the manuscript since this is an important entity which neonatologists are aware of but hasn’t been officially reported. Therefore the subject is both novel and clinically important. However, in my opinion it needs professional scientific editing to be taken seriously by readers. I will lay out my suggestions for the abstract to make my point.

We thank the reviewer for his positive critics and the suggestions which make the manuscript much more readable and synthetized. In fact the title and the abstract are completely re-written by him. Thank you.

1. The title of the manuscript is too long. I suggest something like- Clear amniotic fluid aspiration syndrome: a novel description of an old entity. E 

The title is now as suggested

2. Abstract

The introduction of both the abstract and the manuscript shouldn’t contain any of the study’s’ findings. You should only present what’s known in the literature. 

I would say that the study’s’ aims were: 1. to characterize the risk factors and outcomes associated with Clear Amniotic Fluid Aspiration Syndrome and 2. to compare the outcomes of Clear Amniotic Fluid Aspiration to Meconium Aspriration. 

This sentence “The study’s’ aims were: 1. to characterize the risk factors and outcomes associated with Clear Amniotic Fluid Aspiration Syndrome and 2. to compare the outcomes of Clear Amniotic Fluid Aspiration to Meconium Aspriration” is now in “Background”

Methods- Is not professionally worded. It’s grammatically flawed to begin a sentence with a numerical number. I would suggest beginning traditionally: This was an observational study of….over a 22 year period in a single medical center. Compared were parturient/labor characteristics and neonatal outcomes in cases with suspected Clear Amniotic Fluid Aspiration to cases suspected for Meconium Aspiration. 

Methods has been reworded: “This was an observational study over a 22 year period in a single level 3 medical center. Compared were parturient/labor characteristics and neonatal outcomes in cases with suspected Clear Amniotic Fluid Aspiration to cases suspected for Meconium Aspiration.”

Results

Of 79,620 term deliveries there were 66,705 (83.8%) clear amniotic fluids and 12,915 (16.2%) meconium stained amniotic fluid (MSAF) ; Of neonates born with clear amniotic fluid, 166 (0.25%) were diagnosed with Clear Amniotic Fluid Aspiration Syndrome (CAF-AS) While 202 (15.7%) of those born with MSAF, were diagnosed with aspiration syndrome (MSAF-AS). Both conditions had comparable rates of mild manifestation (% vs%, p=). Persistent pulmonary hypertension (PPH) occurred 5 times less in CAF-AS than MSAF-AS ( OR vs OR, p <) Both conditions presented similar rates of surfactant destruction without PPH (% vs %, p=). There was 1 postnatal death in CAF-AS vs 10 in MSAF-AS.

Results are now: Of 79,620 term deliveries there were 66,705 (83.8%) clear amniotic fluids and 12,915 (16.2%) meconium stained amniotic fluid (MSAF) ; Of neonates born with clear amniotic fluid, 166 (0.25%) were diagnosed with Clear Amniotic Fluid Aspiration Syndrome (CAF-AS) While 202 (15.7%) of those born with MSAF, were diagnosed with aspiration syndrome (MSAF-AS). Both conditions had comparable rates of mild manifestation (67.5% vs 69.2%, p= 0.63). Persistent pulmonary hypertension (PPH) occurred 5 times less in CAF-AS than MSAF-AS (4% vs 20%, OR 0.17, P< 0.0001) Both conditions presented similar rates of surfactant destruction without PPH (11.1 % vs 13.4%, p= 0.87). There was 1 postnatal death in CAF-AS vs 10 in MSAF-AS.

Conclusion

The conclusion now is “CAF-AS were quantitatively quite similar in terms of need of actual active intervention of the neonatologists in the delivery room (166 vs 202, i.e. in terms of numbers of cases and not prevalence) to MSAF-AS We identified in these cases two major specific causes: hyperkinetic explosive deliveries in multiparas and long-lasting episodes of maternal hypotension due to epidural/spinal anaesthesia during labour. Out of 140 million births per year in the world, it should concern 3 million neglected cases nowadays. Future studies should evaluate if this CAF-AS should benefit from a more active intervention such as immediate endotracheal suction at birth, this clear fluid being very easy to suction.”

Reviewer comment: The conclusion states findings that aren’t presented in the results, such as the hyperkinetic explosive deliveries and the maternal hypotension.

They are not yet presented in results but in the discussion. As the entity being supposed not to exist, nobody was obliged to write down any special observation or suggestions in these kind of records

 Also, the first sentence is misleading. It refers to the number of cases in the study, and compares it between the clear amniotic fluid aspiration and meconium aspiration. For a meaningful comparison, you need to compare the prevalence and not the number. If the denominator is the number of term deliveries for both conditions, then they will have a similar prevalence. 

We have then precised now; “CAF-AS were quantitatively quite similar in terms of need of actual active intervention of the neonatologists in the delivery room (166 vs 202, i.e. in terms of numbers of cases and not prevalence) to MSAF-AS.”

I our mind, it is important that neonatologist clinicians (and especially the Junior ones) REALIZE that in terms of actual work (i.e. NUMBERS and not prevalence), they have to take care equally for both entities which was absolutely not the case the last 2 decades.

In results, we show clearly that the CAF-AS are highly underestimated (rate of no intubations and primary intubations while secondary intubations are OR 1.7, p almost significant 0.09). This means that we let these poor babies “rowing” alone with respiratory distress during long hours before being relieved.

Introduction

Like the rest of the manuscript, the introduction lacks scientific professional editing. Here are the main comments:

-The first sentence needs citing of references. 

We have put 8 references now

-The use of parenthesis and punctuation marks to introduce thoughts is not accepted in scientific papers. I refer to the following sentence:

“Hence, MSAF-AS may be probably the topic that induced the 

highest controversies [3] if not errors of judgment (« tales » ?).”

-Use of questions to make a point, is not accepted in scientific papers such as:

“Should we perform endotracheal intubation at birth in the delivery room, and to whom? Should we suction the liquid coming from the lung? Vigorous on non-vigorous babies?”

The paragraph : “Hence, MSAF-AS may be probably the topic that induced the highest controversies [3] if not errors of judgment (« tales » ?) in the short history of the new science “neonatology” (5-6 decades). Should we perform endotracheal intubation at birth in the delivery room, and to whom? Should we suction the liquid coming from the lung? Vigorous on non-vigorous babies? “

Has been completely deleted

-The aim of the study should be clear and concise. Currently the paragraph intended to lay out the aims of the study is a long paragraph that could be summarized to say:

 CAF-AS is a condition never reported before in the literature. We aim to describe associated risk factors and outcome and compare it to meconium aspiration syndrome.

Your last sentence replaces now the original one 

Methods

All the relevant information is present. But scientific editing is significantly needed. Please write the criteria needed for the diagnosis of CAF-AS. Also add the outcomes assessed.

We have added:

The criteria needed for the diagnosis of CAF-AS: severe respiratory distress in the newbornat birth (silverman score > 3), non-vigorous hypoxic baby needing tracheal intubation, abundant tracheal production of amniotic fluid at suction in the delivery room (and careful timing of this production). Bilateral crackling lung sounds .Need of oxygen inducing a transfer in the neonatal department (passive oxygen, n-cpap, intubation). Radiological aspects at chest X ray similar to that of MSAF-AS.

Outcomes assessed.: hours or days of oxygenation, hours or days of n-cpap or intubated mechanical ventilation with oxygen, destruction of surfactant, persistent pulmonary hypertension, death.

Statistical analysis section is well written.

Ethical approval section is too long and cumbersome. 

It is now “This study was conducted in accordance with French legislation. As per new French law applicable to trials involving human subjects, a specific approval of an ethics committee is not required for this non-interventional study with anonymized data” .

Results

Scientific editing is required for the text and the tables. The statistics seem fine. I am not familiar with the term “explosive delivery” that appears in the conclusion, explosive has been replaced by rapid

 nor do I see this or maternal hypotension, in the results section, both of which appear in the conclusion. You can’t state in the conclusion findings that do not appear in the results

We have added a paragraph in results: Clinical conditions associated with CAF-AS. The entity CAF-AS being supposed not to exist, nobody was obliged to write down any special observation or suggestions in these kind of records. Therefore, in this paragraph that we will develop inthe discussion. In these non-protocoled observations, we may recall three main entiteis : First, what we may call “acute events” at birth such as difficult and long extraction (forceps, vacuum, spatulas), prolonged foetal bradycardia (10 mn), shoulder dystocia, abruption placenta, maternal septic shock, foetal acute anaemia. the two other features which seem to be very specific to the CAF-AS are, second, hyperkinetic rapid deliveries in multiparas (“champagne cork” deliveries) and, third, long-lasting (8-10 minutes) episodes of maternal hypotension due to epidural/spinal anaesthesia (occurring even 30 minutes before expulsion).

Discussion

Scientific editing is heavily required. What exactly is the point of figure 1? The discussion usually doesn’t have figures or tables. 

We have put all the Figures (chest X-Rays) in results, letting the Editor to choose where he will put them in the final article

To sum up, the manuscript addresses a novel subject. The methods and analyses are fine. However, the manuscript requires heavy scientific editing to become publishable. 

All the manuscript has been revised by a CAMBRIGE PROOFREADING expert

---

## [Decision Letter · Decision Letter 2]

20 Feb 2024

PONE-D-23-31689R2Clear amniotic fluid aspiration syndrome: a novel description of an old entity.PLOS ONE

Dear Dr. Robillard,

Thank you for submitting your manuscript to PLOS ONE. After careful consideration, we feel that it has merit but does not fully meet PLOS ONE’s publication criteria as it currently stands. Therefore, we invite you to submit a revised version of the manuscript that addresses the points raised during the review process.

**ACADEMIC EDITOR: Please respond to all reviewers comments** 

We look forward to receiving your revised manuscript.

Kind regards,

Ahmed Mohamed Maged, MD

Academic Editor

PLOS ONE

Reviewers' comments:

Reviewer's Responses to Questions

**Comments to the Author**

1. If the authors have adequately addressed your comments raised in a previous round of review and you feel that this manuscript is now acceptable for publication, you may indicate that here to bypass the “Comments to the Author” section, enter your conflict of interest statement in the “Confidential to Editor” section, and submit your "Accept" recommendation.

Reviewer #1: (No Response)

Reviewer #4: All comments have been addressed

2. Is the manuscript technically sound, and do the data support the conclusions?

Reviewer #1: No

Reviewer #4: No

3. Has the statistical analysis been performed appropriately and rigorously? 

Reviewer #1: N/A

Reviewer #4: No

4. Have the authors made all data underlying the findings in their manuscript fully available?

Reviewer #1: No

Reviewer #4: Yes

5. Is the manuscript presented in an intelligible fashion and written in standard English?

Reviewer #1: No

Reviewer #4: No

6. Review Comments to the Author

Reviewer #1: Dear Authors I am very sorry but the modified version of your paper still remain not sufficient to be published, above all in a journal with a high impact factor

Reviewer #4: Abstract

What is surfactant destruction and how was this diagnosed?

Please display the percentage for postnatal death, currently you state the absolute number 1 vs 10.

The conclusion is unacceptable in its current form. It is not supported by the results presented in the abstract. It introduces new findings not mentioned in the results. Also, its sentences should be more concise, laying out insights with minimal use of words.

There are not enough keywords. “Suction” is nonspecific, use the commonly used term to make it clear that its endotracheal suction. I am not a neonatologist, so look up the accepted term.

Main Text

Introduction

Poorly written. This is not clean scientific language which is more objective and neutral. Currently, it has a lot of opinions and descriptive language in it, portraying the feelings of the author.

Methods

There is no methods section. The study design is not described. The definitions are a good start. Make sure to define every term needed. The definitions should have a reference if they are internationally accepted.

Results

The authors introduce terms that need clarification, like- primary/secondary intubation.

The language used is still non-scientific, for example- : “neonates were kept for dozens of minutes (even hours)”… an accurate mean time should be presented instead of a rough verbal estimation.

Or “CAF-AS were rather a problem of multiparous deliveries” instead of writing that there was a higher prevalence of multiparity in the CAFS group compared to the MAS group….

The figures are not clear If they are a description of the different cases, then present them in a table. Figures are usually a graphic description or image, not a list of case descriptions.

In the results you display the number of minutes the babies were suctioned, but in the methods you do not detail that suctioning was timed in all cases.

There are many more examples of flawed methodology and unprofessional scientific writing.

Discussion

The discussion is mostly based on the author’s experience and not on scientific evidence.

The limitation of imperfect study methodology and unprofessional scientific writing is prominent in this section as it was in previous sections.

I would suggest the authors to reorganize the data, improve the display of data, make sure every parameter is well defined and then get assistance on scientific writing, to maximize the potential of this manuscript. In my opinion, it is unpublishable in its current form. I hope the authors will succeed in doing this and getting this paper published because the subject itself seems very important clinically.

7. PLOS authors have the option to publish the peer review history of their article (what does this mean?). If published, this will include your full peer review and any attached files.

Reviewer #1: No

Reviewer #4: No

---

## [Author Response · Author response to Decision Letter 2]

6 Mar 2024

We thank the reviewer for his positive critics and the suggestions which make the manuscript much more readable and synthetized. Modifications are in red in the text

Abstract

What is surfactant destruction and how was this diagnosed?

You are right, destruction of surfactant is an interpretation. We replaced destruction of surfactant by “need of surfactant” in all the paper. In meconium aspiration syndrome, it has been recommended for 2 decades to try surfactant therapy in these babies although they are all at term (and it works). Neonatologists interpret this as “destruction of surfactant” or “disorientation of surfactant” due to the mechanical aggression of the inhaled fluid.

Please display the percentage for postnatal death, currently you state the absolute number 1 vs 10.

This has been done and the odds-ratios calculated

The conclusion is unacceptable in its current form. It is not supported by the results presented in the abstract. It introduces new findings not mentioned in the results.

Also, its sentences should be more concise, laying out insights with minimal use of words.

The conclusion now is: “CAF-AS were quantitatively quite similar in terms of need of actual active intervention of the neonatologists in the delivery room (166 vs 202, i.e. in terms of numbers of cases and not prevalence) to MSAF-AS We identified in these cases two major specific causes: hyperkinetic explosive deliveries in multiparas and long-lasting episodes of maternal hypotension due to epidural/spinal anaesthesia during labour. Out of 140 million births per year in the world, it should concern 3 million neglected cases nowadays. Future studies should evaluate if this CAF-AS should benefit from a more active intervention such as immediate endotracheal suction at birth, this clear fluid being very easy to suction.”

There are not enough keywords. “Suction” is nonspecific,

Keywords are now: Meconium aspiration syndrome, clear amniotic fluid aspiration syndrome, endotracheal suction

 use the commonly used term to make it clear that its endotracheal suction. I am not a neonatologist, so look up the accepted term. 

We changed everywhere in the text the term “suction” by endotracheal suction

Main Text

Introduction

Poorly written. This is not clean scientific language which is more objective and neutral. Currently, it has a lot of opinions and descriptive language in it, portraying the feelings of the author.

The intoduction is now:

Meconium-stained amniotic fluid aspiration syndrome MSAF-AS has been particularly well reported for 5 decades as it may lead in newborns to severe respiratory distress syndrome with a high morbidity and mortality rate [1-7}. The terror induced by this syndrome is precisely magnified by the fact that these severe features occur in term neonates. A recent paper by Huang, Zhu and Cheng [7] comparing practices between China and the USA in the past 60 years is probably one of the best recalling exhaustively decades all the debates concerning management of MSAF-AS. International recommendations are very well synthetized in a table. These authors summarize all the neonatal resuscitation’s international recommendations: 1987, 1994, 2000, 2005-2006, 2011, 2016, and 2020 (some of them contradicting the preceding). Monfredini et al. [2] also recall very well all these evolutions and controversies [2]. 

“Clear amniotic fluid aspiration syndrome” CAF-AS does not yet exist in PubMed search. Typing such research gives 56 results ALL of them speaking of meconium staining (because of the root “amniotic fluid”) except one: “Suctioning of clear amniotic fluid at birth: A systematic review” [8], but the authors speak only of suctioning the mouth and nose at birth. If we type “meconium-stained amniotic fluid”, we have 1188 results. 

CAF-AS is a condition never reported before in the literature. We aim to describe associated risk factors and outcomes and compare them to meconium aspiration syndrome. 

Methods

There is no methods section. The study design is not described. The definitions are a good start. Make sure to define every term needed. The definitions should have a reference if they are internationally accepted. The reference internationally accepted is reference 3

We have added:

“The criteria needed for the diagnosis of CAF-AS: severe respiratory distress in the newbornat birth (silverman score > 3), non-vigorous hypoxic baby needing tracheal intubation, abundant tracheal production of amniotic fluid at suction in the delivery room (and careful timing of this production). Bilateral crackling lung sounds .Need of oxygen inducing a transfer in the neonatal department (passive oxygen, n-cpap, intubation). Radiological aspects at chest X ray similar to that of MSAF-AS.

Outcomes assessed.: hours or days of oxygenation, hours or days of n-cpap or intubated mechanical ventilation with oxygen, destruction of surfactant, persistent pulmonary hypertension, death.”

Results

The authors introduce terms that need clarification, like- primary/secondary intubation. 

We have added in methods (page 4): “Primary intubation means intubation at birth directly in the delivery room. Secondary intubation means that the newborn has been intubated in the neonatal NICU department after transfer, therefore minutes or hours after birth”

The language used is still non-scientific, for example- : “neonates were kept for dozens of minutes (even hours)”… an accurate mean time should be presented instead of a rough verbal estimation.

We rephrased the sentence by “newborns with respiratory distress syndromes after a long stay at the maternity ward (even hours) before a transfer decision in the neonatal department.”, page 5

Or “CAF-AS were rather a problem of multiparous deliveries” instead of writing that there was a higher prevalence of multiparity in the CAFS group compared to the MAS group….

We have replaced (page 5) by your suggested sentence. “It is of note that there was a higher prevalence of multiparity in the CAF-AS group (while it has long been known that the MAS-AS is largely associated with primiparities), p= 0.01.” 

The figures are not clear If they are a description of the different cases, then present them in a table. Figures are usually a graphic description or image, not a list of case descriptions.

Again, with all my respect, I think that all these chest X-rays are indispensable to CONVINCE NEONATOLOGISTS. They show (at a lesser extent) typical images encountered in the meconium aspiration syndrome that neonatologists know by heart. Without these, 80% of them will continue to ignore this syndrome.

In the results you display the number of minutes the babies were suctioned, but in the methods you do not detail that suctioning was timed in all cases.

In the methods, we have added (page 4).” All pediatrical interventions in the delivery room (intubation or not) were systematically timed by initializing the infant resuscitation table’s chronometer to evaluate the duration of intervention and/or the timing of intubation”

There are many more examples of flawed methodology (please see the first paragraph of the chapter below) and unprofessional scientific writing. 

Discussion

The discussion is mostly based on the author’s experience and not on scientific evidence. 

- With all my respect, it could not be another way, because IT IS YET AN UNKNOWN SYDROME. Therefore, it was not possible to write a specific protocol for an inexistent disease. I just copy the comment of reviewer 3: “General comment- I am very enthusiastic for the manuscript since this is an important entity which neonatologists are aware of but hasn’t been officially reported. Therefore the subject is both novel and clinically important.”

- 1) We have added in the introduction “CAF-AS is a condition never reported before in the literature. We aim to describe associated risk factors and outcome and compare it to meconium aspiration syndrome.”, page 3

- 2) We have added a paragraph in results: “Clinical conditions associated with CAF-AS. The entity CAF-AS being supposed not to exist, nobody was obliged to write down any special observation or suggestions in these kind of records. Therefore, in this paragraph that we will develop inthe discussion. In these non-protocoled observations, we may recall three main entities : First, what we may call “acute events” at birth such as difficult and long extraction (forceps, vacuum, spatulas), prolonged foetal bradycardia (10 mn), shoulder dystocia, abruption placenta, maternal septic shock, foetal acute anaemia. the two other features which seem to be very specific to the CAF-AS are, second, hyperkinetic rapid deliveries in multiparas (“champagne cork” deliveries) and, third, long-lasting (8-10 minutes) episodes of maternal hypotension due to epidural/spinal anaesthesia (occurring even 30 minutes before expulsion).”

- 

The limitation of imperfect study methodology and unprofessional scientific writing is prominent in this section as it was in previous sections.

We have asked to a CAMBRIGE PROOFREADING expert

to revise the entire manuscript 

I would suggest the authors to reorganize the data, improve the display of data, make sure every parameter is well defined and then get assistance on scientific writing, to maximize the potential of this manuscript. In my opinion, it is unpublishable in its current form. I hope the authors will succeed in doing this and getting this paper published because the subject itself seems very important clinically.

---

## [Editor Report · Decision Letter 3]

19 Mar 2024

Clear amniotic fluid aspiration syndrome: a novel description of an old entity.

PONE-D-23-31689R3

Dear Dr. Robillard,

We’re pleased to inform you that your manuscript has been judged scientifically suitable for publication and will be formally accepted for publication once it meets all outstanding technical requirements.

Kind regards,

Ahmed Mohamed Maged, MD

Academic Editor

PLOS ONE

Additional Editor Comments (optional):

All reviewers comments has been addressed
---

## [Editor Report · Acceptance letter]

26 Mar 2024

PONE-D-23-31689R3 

PLOS ONE

Dear Dr. Robillard, 

I'm pleased to inform you that your manuscript has been deemed suitable for publication in PLOS ONE. Congratulations! Your manuscript is now being handed over to our production team.

Kind regards, 

on behalf of

Professor Ahmed Mohamed Maged 

Academic Editor

PLOS ONE